# Detecting Anomalies within Time Series using Local Neural Transformations

## Abstract

We develop a new method to detect anomalies within time series, which is essential in many application domains, reaching from self-driving cars, finance, and marketing to medical diagnosis and epidemiology. The method is based on self-supervised deep learning that has played a key role in facilitating deep anomaly detection on images, where powerful image transformations are available. However, such transformations are widely unavailable for time series. Addressing this, we develop Local Neural Transformations (LNT), a method learning local transformations of time series from data. The method produces an anomaly score for each time step and thus can be used to detect anomalies within time series. We prove in a theoretical analysis that our novel training objective is more suitable for transformation learning than previous deep Anomaly detection (AD) methods. Our experiments demonstrate that LNT can find anomalies in speech segments from the LibriSpeech data set and better detect interruptions to cyber-physical systems than previous work. Visualization of the learned transformations gives insight into the type of transformations that LNT learns.

## 1 Introduction

Anomaly detection (AD) in time series is significant in many industrial, medical, and scientific applications. For instance, undetected anomalies in water treatment facilities or chemical plants can bring harm to millions of people. Such systems need to be constantly monitored for anomalies.

While AD has been an important field in machine learning for several decades (Ruff et al., 2020), promising performance gains have been primarily reported in applying deep learning methods to high-dimensional data such as images (Golan & El-Yaniv, 2018; Wang et al., 2019; Hendrycks et al., 2019; Bergman & Hoshen, 2020). Time series exhibit complex temporal dependencies and can be even more diverse than natural images. Consequently, time series anomaly detection with deep learning approaches has been widely studied in recent years Zhou et al. (2019); Shen et al. (2020); Malhotra et al. (2016); Li et al. (2019); de Haan & Löwe (2021); Deng & Hooi (2021); Carmona et al. (2021). While *unsupervised* methods based on density estimation can yield poor results for AD (Nalisnick et al., 2018), a recent trend relying on *self-supervision* has proven superior performance. As detailed below, this paper attempts to integrate recent ideas from self-supervised AD of non-temporal data with modern deep learning architectures for sequence modeling.

In this line of work, one uses auxiliary tasks, often based on data augmentation, both for training and anomaly scoring. Data augmentation usually relies on hand-designed data transformations such as rotations for images (Golan & El-Yaniv, 2018; Wang et al., 2019; Hendrycks et al., 2019). Qiu et al. (2021) showed that these transformations could instead be *learned*, thereby making self-supervised AD applicable to specialized domains beyond images. While this approach can identify an entire sequence as anomalous, it can still not be applied to detecting anomalies *within* time series (i.e., on a sub-sequence level).

But this adaption is not straightforward: For AD within time series, both local semantics (the dynamics within a time window) and contextualized semantics (how the time window relates to the remaining time series) matter. To capture both, we propose an end-to-end approach that combines time series representations (Oord et al., 2018) with a novel transformation learning objective. As a result, the local transformations create

different views of the data in the latent space (Rudolph et al., 2017) (as opposed to applying them to the data directly as in Qiu et al. (2021)).

We develop Local Neural Transformations (LNT): a novel objective that combines representation learning with transformation learning. The encoder for feature extraction and the neural transformations are trained jointly on this loss. We show that the learned latent transformations can correspond to interpretable effects: in one experiment on speech data (details in Section 5), LNT learns transformations that insert delays. Neural transformations are much more general than hand-crafted transformations, which for time series could be time warping, reflections, or shifts: as we illustrate, they can transform the data in ways unintuitive to humans but valuable for the downstream task of AD.

We prove theoretically (Section 4) and show empirically (Section 5) that combining representation and transformation learning is beneficial for detecting anomalies within time series. LNT outperforms various AD techniques on benchmark data, including a baseline using the Contrastive Predictive Coding (CPC) loss as the anomaly score (de Haan & Löwe, 2021). We evaluate the methods on public AD datasets for time series from cyber-physical systems. Furthermore, we detect artificial anomalies in speech data, which is challenging due to its complex temporal dynamics. In some experiments, LNT outperforms many strong baselines.

To summarize, our contributions in this work are:

1. A *new method*, LNT, for AD within time series. It unifies time series representations with a novel approach for learning *local* transformations. A open-source pytorch implementation is available at []¹.

2. A *theoretical analysis*. We prove that both learning paradigms complement each other to avoid trivial solutions not appropriate for detecting anomalies.

3. An *empirical study* showing that LNT can detect anomalies within real cyber-physical data streams on par or better than many existing methods.

## 2   Related Work

We first describe related work in time series AD, which is the problem we tackle in this work. We then describe related methods, specifically advances in self-supervised AD.

### 2.1   Time series anomaly detection

There are two types of anomalies in time series: local and global anomalies. Global anomalies are entire time series, with a single anomaly score for the entire series. Local anomalies occur at isolated timestamps or short time intervals within the time series, so each time point must be assigned with an anomaly score. This is the setting that we consider in this work. Existing methods for local AD in time series using deep learning can be divided into four categories, discussed in detail below: (i) methods based on sequence forecasting, (ii) autoencoders, (iii) generative sequence models, and (iv) other approaches.

**Forecasting methods**   A straightforward approach to detect anomalies in time series is to use the error of a time-series forecaster (predicting the value of the next time step from the time series' past history) as an anomaly score. The rationale behind is that a forecaster trained on mostly normal data will err less on normal than on abnormal data. We may use any time-series regression method as the forecaster, and various methods have been studied, including neural architectures such as recurrent neural networks (RNNs) (Malhotra et al., 2015; Filonov et al., 2016) and temporal convolutional neural networks (TCNs) (He & Zhao, 2019; Munir et al., 2019), where the convolution operation is applied along the temporal dimension only.

**Autoencoders**   To detect anomalies within time series, AEs have been combined with various neural network architectures, including RNNs (Malhotra et al., 2016) and TCNs (Thill et al., 2020) or variants (Zhang et al., 2019). Audibert et al. (2020) propose an architecture based purely on dense layers using a

---

¹link removed to preserve anonymity

combination of two AEs connected with the adversarial loss. Again, the rational of using such approaches for AD is that after training on normal data, a high reconstruction error can be used to detect anomalies.

**Deep generative models**  Variational autoencoders (VAEs) (Kingma & Welling, 2014) have frequently been combined with RNNs (Sölch et al., 2016; Park et al., 2018) to detect anomalies within time series. Pereira & Silveira (2018) combine an RNN with temporal self-attention. Guo et al. (2018) use gated recurrent units (GRUs) in combination with a gaussian mixture model. Su et al. (2019) augment a GRU-based VAE with a normalizing flow and a linear Gaussian state-space model. Generative adversarial networks (Goodfellow et al., 2014) have been used for AD within time series, taking either the discriminator's error (Liang et al., 2021) or the generator's residuals (Zhou et al., 2019) as an anomaly score. Li et al. (2019) use a weighted combination of both. These approaches have been combined with TCNs (Zhou et al., 2019) and RNNs (Niu et al., 2020; Geiger et al., 2020).

**Other methods**  Some of the above-described approaches have been used in combination. For instance, Zhao et al. (2020) combine TCNs and LSTMs. Shen et al. (2020) combine a dilated RNN with a deep multi-sphere hypersphere classifier on the cluster centers of a hierarchical clustering procedure, with regularizers encouraging orthogonal centers at each layer and prediction regularizers encouraging useful representations in intermediate layers. Deng & Hooi (2021) construct a graph with nodes for each feature and edges representing relations between features; these are learned and combined with a graph-based attention mechanism. Carmona et al. (2021) employ a TCN as an encoder to train a hypersphere classifier in the latent space, with the option of including known anomalies into training.

## 2.2  Self-supervised anomaly detection

Recently, there has been growing interest in tackling AD with *self-supervised learning*. The core idea of self-supervised learning is to devise training tasks, often based on data augmentation, that guide the model to learn useful representations of the data. In self-supervised AD, performance on the auxiliary tasks can be used for anomaly scoring. This is justified by the principle of inlier priority (Wang et al., 2019) which posits that a self-supervised approach will prioritize solving its training task for inliers. End-to-end detection methods based on transformation prediction (Golan & El-Yaniv, 2018; Hendrycks et al., 2019) have been designed for image AD. However, they require effective hand-crafted transformations while for data types beyond images, it is hard to design effective transformations by hand. Previous works proposed to utilize random affine transformations (Bergman & Hoshen, 2020) or data-driven neural transformations (Qiu et al., 2021) for AD. Neural transformations have been used to detect entire anomalous sequences. However, when the neural transformation learning approach of Qiu et al. (2021) is applied to the task of local anomaly detection, it can lead to trivial transformations that are not suitable for AD. Our work proves this and introduces a novel *local* transformation learning objective.

Alternatively, de Haan & Löwe (2021) propose to use the training criterion of CPC, a self-supervised approach without data augmentation, for anomaly detection. CPC learns local time series representations via contrastive predictions of future representations (Oord et al., 2018). However, the CPC loss is not a good fit for scoring anomalies since it requires a random draw of negative samples, which leads to a biased estimation or high memory cost during test time (de Haan & Löwe, 2021). Our work overcomes this.

# 3  Method

In this work, we propose *Local Neural Transformations (LNT)*, a new framework for detecting anomalies within time series data. LNT has two components: feature extraction and feature transformations. Given an input sequence, an encoder produces an embedding for each time step, encoding relevant information from the current time window. These features are then transformed by applying distinct neural networks to each embedding, producing different *latent views*. The views are trained to fulfill two requirements; the views should be diverse and semantically meaningful, i.e., they should reflect both local dynamics as well as how the observations fit into the larger context of the time series. Both are encouraged via self-supervision.

Specifically, two aspects of LNT are *self-supervised*: it combines two different contrastive losses. One of the contrastive losses, CPC, guides the representation learning that guarantees the encoder of LNT to produce good semantic time series representations that generalize well to unseen test data. The second contrastive loss, a novel dynamic deterministic contrastive loss (DDCL), contrasts different latent views of each time step to encourage the latent views to be diverse and semantically representative of the time series, both in a local and in a contextualized sense.

LNT follows the general paradigm of self-supervised AD. During training, the capability to contrast the data views produced by the transformations improves for the normal data, while it deteriorates for anomalies. The main components of LNT are the encoder producing local representations of the input and local neural transformations, which are neural networks that transform the local representations into different views. The encoder and the local neural transformations are trained using the two losses, one guiding the quality of representations, the other guiding the quality of the transformations. The losses are combined to produce an anomaly score $\ell_t$ for each time step in the input time series $x_{1:t} := (x_1, \ldots, x_t)^T$ ; $x_t \in \mathbb{R}^d$, representing the likelihood that the observation in this time step is an anomaly. Formally, we assume a time series $x_{1:t}$ to be observations of random variables $X_{1:t}$. Further, we assume that its *data generating distribution* factorizes with context variables as detailed in the definition below.

**Definition 1** (Temporal Anomaly). *Let $x_{1:t} := (x_1, \ldots, x_t)^T \in \mathbb{R}^{T \times d}$ be a multivariate time series and $p$ a distribution that factorizes with context variables $C_{1:t}$ as $p(x_{1:T}) = \int \prod_t p(X_t = x_t | C_t) p(C_t | C_{t-1}) dC_{1:t}$ and $x_{1:t} \sim p$. We call an observation $\tilde{x}_t$ an anomaly iff $\tilde{x}_t \nsim p(x_t | x_{1:t-1})$, i.e. it is sampled from a different data generating distribution.*

Given time series data $\{x_{1:t}\}$ it is unclear how to choose a good context representation $C_{1:t}$ that is able to explain all the effects like trends, seasonality, or change points in a time series. This aspect renders it a challenging representation learning problem. Also, good context variables $C_t$ might evolve on a coarse time scale than the measurements recorded for a time series. Note that from definition 1 not every change point is necessarily an anomaly as long as similar changes have been sufficiently observed during training and are accounted for in $p(C_t | C_{t-1})$. In that way a time series with non-stationary dynamics and many change points (compare *LibriSpeech*, section 5.1) can be normal.

Before presenting local transformation learning and the DDCL in Section 3.2, we will first describe the encoder and the CPC-loss in Section 3.1. Then, we discuss how a trained model is used to detect anomalies. Finally, in Section 4, we provide theoretical arguments for combining transformation learning with representation learning. All notations used throughout the remainder of the section are summarized in table 4.

## 3.1 Local Time Series Representations

The LNT architecture has two components, a feature extractor (encoder) and an anomaly detector (local neural transformations). The encoder maps a sequence of samples to a sequence of local latent representations $z_t$ and is trained using the principles of *Contrastive Predictive Coding (CPC)* (Oord et al., 2018). We use the same architecture as Oord et al. (2018). The representations produced by the encoder $z_t = g_{\text{enc}}(x_t)$ are summarized with an autoregressive module into context vectors $c_t = g_{\text{ar}}(z_{\leq t})$. For different choices of $t$ and prediction steps $k$, we built mini batches by randomly sampling a set $X$ of size $N$ from the training data that each contains one positive pair $(x_t, x_{t+k})$ and $N - 1$ negative pairs $(x_t, x_j)$, with $x_j$ being any other sample $(j \neq t + k)$ from the same mini batch but for a different choice of $t$ and $k$. The CPC loss contrasts linear $k$-step future predictions $W_k c_t$ against negative samples:

$$\mathcal{L}_{\text{CPC}} = -\mathbb{E}_{X \sim \mathcal{D}} \left[ \log \frac{\exp(z_{t+k}^T W_k c_t)}{\sum_X \exp(z_j^T W_k c_t)} \right]. \tag{1}$$

It encourages the context representation $c_t$ to be predictive of nearby local representations $z_{t+k}$. Optimizing Equation (1) relates to maximizing the mutual information (Tschannen et al., 2019) between the context representation $c_t$ and nearby time points $x_{t+k}$ to produce good representations ($z_t$ and $c_t$) that can be used in downstream tasks, including AD.

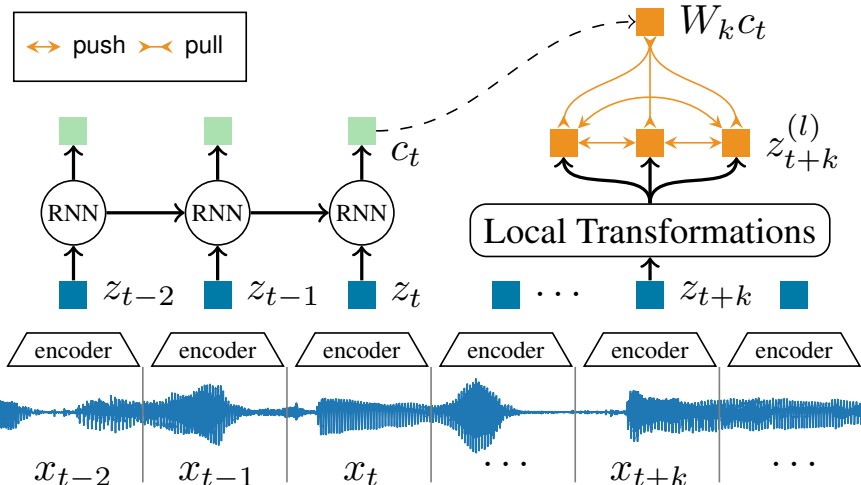

Figure 1: LNT on latent representations $z_t$ resulting in transformed views $\mathcal{T}_l(z_t)$ - it can be viewed as pushing and pulling representations in latent space with the Dynamic Deterministic Contrastive Loss (DDCL)

### 3.2 Local Neural Transformations

The second part of the LNT architecture introduces an auxiliary task for AD. The time series representations $z_t$ are processed by local neural transformations to produce different views of each embedding. This operation relates to data augmentation but has two major differences: First, the transformations are not applied at the data level but in the latent space, producing *latent views* of each time window. Second, the transformations are not hand-crafted as is often done in computer vision, where rotation, cropping and blurring are popular augmentations, but are instead directly learned during training (Tamkin et al., 2020; Qiu et al., 2021).

The neural transformations are $L$ neural networks $\mathcal{T}_l(\cdot)$ with parameters $\theta_l$. They are applied to each latent representation $z_t$ to produce different latent views $z_t^{(l)} = \mathcal{T}_l(z_t)$, as shown in Figure 1. Each of the transformed views is encouraged to be predictive of the context at different time horizons $k$ by a loss contribution

$$\ell_t^{(k,l)}(x_{\leq t}) = -\log \frac{h\big(z_t^{(l)}, W_k c_{t-k}\big)}{h\big(z_t^{(l)}, W_k c_{t-k}\big) + \sum\limits_{m \neq l} h\big(z_t^{(l)}, z_t^{(m)}\big)}, \tag{2}$$

which simultaneously pushes different views of the same latent representations apart from each other. The notation $h(z_i, z_j) := \exp \frac{z_i^T z_j}{\|z_i\|\|z_j\|}$ is defined as the exponentiated cosine similarity in the embedding space. Unlike most contrastive losses, where the negative samples are drawn from a noise distribution (Gutmann & Hyvärinen, 2012), the other views to contrast against are constructed deterministically from the same input (Qiu et al., 2021). The loss contributions of each time-step $t$, each transformation $l$, and each time horizon $k$ are combined to produce the *Dynamic Deterministic Contrastive Loss* (DDCL):

$$\mathcal{L}_{\text{DDCL}} = \mathbb{E}_{x_{1:T} \sim \mathcal{D}} \left[ \sum_{k=1}^{K} \sum_{t=1}^{T} \sum_{l=1}^{L} \ell_t^{(k,l)}(x_{\leq t}) \right]. \tag{3}$$

During training, the two objectives (Equations (1) and (3)) are optimized jointly using a unified loss,

$$\mathcal{L} = \mathcal{L}_{\text{CPC}} + \lambda \cdot \mathcal{L}_{\text{DDCL}} \tag{4}$$

and a balancing hyperparameter $\lambda$. All contrasting operations are performed on the mini batch described before, but the deterministic contrasting of distinct transformations $m \neq l$ in DDCL causes the mini batch of latent representations to grow by a factor of $L$. Since for each $z_t$ a set $\{z_t^{(}0), \ldots, z_t^{(}L)\}$ of $L$ distinct views needs to be stored.

As depicted by orange arrows in Figure 1, $\mathcal{L}_{\text{DDCL}}$ can intuitively be interpreted as pushing and pulling different representations in latent space. The numerator pulls the learned transformations $z_{t+k}^{(l)}$ close to $W_k c_t$ ensuring semantic views, while the denominator pushes different views apart, ensuring diversity in the learned transformations.

### 3.2.1 Scoring of Anomalies

After training LNT on a dataset of typical time series, we can use the DDCL for AD. Given a test sequence $x_{1:T}$, we evaluate the contribution of individual time steps to $\mathcal{L}_{\text{DDCL}}$ (Equation (3)). The score for each time point $t$ in the sequence is,

$$\ell_t(x_{\leq t}) = \sum_{k=1}^{K} \sum_{l=1}^{L} \ell_t^{(k,l)}(x_{\leq t}) \tag{5}$$

The higher the score, the more likely the series exhibits abnormal behavior at time $t$. Unlike CPC-based AD (de Haan & Löwe, 2021), this anomaly score has the advantage of being *deterministic* and thus there is no need to draw negative samples from a proposal or noise distribution.

## 4 Analysis

Our experiments in Section 5.5 show that LNT empirically outperforms CPC on various AD tasks. However, since the LNT architecture is trained on two losses jointly (the DDCL and CPC losses), the natural question arises: are *both* losses necessary or could we just train on the DDCL loss alone? The following analysis demonstrates the value of considering both losses jointly.

### 4.1 Ablation Analysis

The following theorem shows that, if we trained the LNT architecture (i.e. the encoder and transformations $\mathcal{T}_i$) only on the $\mathcal{L}_{\text{DDCL}}$ loss (without the $\mathcal{L}_{\text{CPC}}$ loss), the optimal solution would collapse to a constant encoder, a phenomenon known as the *manifold collapse* in deep AD (Ruff et al., 2018). Thus the CPC loss acts as a regularizer in our DDCL framework to avoid the manifold collapse; it is thus strictly necessary.

**Theorem 1.** *Let $g_{enc}^{\theta}$ and $g_{ar}^{\theta}$ be arbitrary encoders (including biases) with learned parameters $\theta$, and let $\mathcal{L}_{DDCL}^{\theta}$ be the corresponding DDCL loss. Then there exist constant encoders $g_{enc}^{\tilde{\theta}}$ and $g_{ar}^{\tilde{\theta}}$ (i.e., $\exists \tilde{\theta}, a, b \; \forall x, z :$ $g_{enc}^{\tilde{\theta}}(x) = a, g_{ar}^{\tilde{\theta}}(z) = b$) with*

$$\mathcal{L}_{DDCL}^{\tilde{\theta}} \leq \mathcal{L}_{DDCL}^{\theta}.$$

*Proof.* Let $g_{enc}^{\theta}$ and $g_{ar}^{\theta}$ be arbitrary encoders (including biases) with learned parameters $\theta$ (for notational simplicity of the proof we understand the additional parameter $W$ as included into $\theta$), and let $\mathcal{L}_{\text{DDCL}}^{\theta}$ be the corresponding DDCL loss. We observe from Equation (3) that $\mathcal{L}_{\text{DDCL}}^{\theta}$ decomposes into a sum of loss contributions $\ell_t^{(k,l)}(x_{\leq t}; \theta)$. Let

$$(x_{\leq t^*}^*, k^*, t^*) = \arg\min \sum_{l=1}^{L} \ell_t^{(k,l)}(x_{\leq t}; \theta), \tag{6}$$

be the indices of the summands with the smallest contribution to the sum, for a given fixed $\theta$. This means $x^*$ is the sample, $k^*$ the time horizon, and $t^*$ the time point associated with the smallest loss contribution to $\mathcal{L}_{\text{DDCL}}$. Put

$$\ell^* := \sum_{l=1}^{L} \ell_t^{(k^*,l)}(x_{\leq t^*}; \theta). \tag{7}$$

Since our encoders are equipped with bias terms there exist constant encoders $g_{enc}^{\tilde{\theta}}$ and $g_{ar}^{\tilde{\theta}}$ (i.e., $\exists \tilde{\theta}, a, b \forall x, z : g_{enc}^{\tilde{\theta}}(x) = a, g_{ar}^{\tilde{\theta}}(z) = b$) with

$$\forall x, k, t : \sum_{l=1}^{L} \ell_t^{(k,l)}(x_{\leq t}; \tilde{\theta}) = \ell^*. \tag{8}$$

Then we have:

$$\mathcal{L}_{\text{DDCL}}(\theta) \overset{equation\ 3}{=} \mathbb{E}\left[ \sum_{k=1}^{K} \sum_{t}^{T} \sum_{l=1}^{L} \ell_t^{(k,l)}(x_{\leq t}; \theta) \right]$$

$$\overset{equation\ 6}{\geq} KT \sum_{l=1}^{L} \ell_t^{(k^*,l)}(x_{\leq t^*}^*; \theta) \overset{equation\ 7}{=} KT\ell^*$$

$$\overset{equation\ 8}{=} \mathbb{E}\left[ \sum_{k=1}^{K} \sum_{t}^{T} \sum_{l=1}^{L} \ell_t^{(k,l)}(x_{\leq t}; \tilde{\theta}) \right] \overset{equation\ 3}{=} \mathcal{L}_{\text{DDCL}}(\tilde{\theta}),$$

which was to prove. □

The above theorem shows that if LNT was trained on the DDCL loss only, LNT would collapse into a trivial solution. On the other hand a constant encoder clearly does not optimize the maximum mutual information criterion (Oord et al., 2018), which is induced by the CPC objective.

## 4.2 Computational Complexity Analysis

This section is dedicated to investigating the computational complexity for both *training* and *scoring anomalies* of the LNT algorithm. A good proxy for complexity is counting the inner product occurring in the loss functions (eqs. (1) and (2)), since each inner product corresponds to an acquisition of the embeddings $z_t, c_t, z_t^{(l)}$ (a forward pass through a fixed network in $\mathcal{O}(1)$) followed by the actual inner product which can be computed in approximately constant time on modern vectorized hardware. Let $B$ denote the batch size in the *training* of LNT. For the CPC part of the loss $\mathcal{O}(K^2 B^2)$ inner products are computed since every embedding $z_t$ in the batch is contrasted against the negatives from a different time series in the same minibatch. For the DDCL part, $\mathcal{O}(BKL^2)$ inner products are required since in eq. (3) every $z_t^{(l)}$ is treated as the positive sample once and contrasted against $L - 1$ negative samples $z_t^{(m)}; m \neq l$, and in practice $K, L \ll B$ are small constants. Thus, in total $\mathcal{O}(B^2 K^2 + BKL^2) = \mathcal{O}(B^2)$. For the complexity of *scoring anomalies*, assume a time series of length $T$. To score $\ell_t$ for a single time step, $\mathcal{O}(KL^2)$ inner products are required. For an entire time series this yields $\mathcal{O}(TL^2)$.

Besides this hard mathematical evidence, there are also other good reasons to include the CPC loss into LNT. For instance, it ensures that the latent representations account for dynamics at longer time scales. This task is carried out by CPC's autoregressive module. Our hypothesis is that, for effective AD within time series, it is necessary to consider both: the local signal in a time window and the larger context across time windows. Otherwise, the observations within a time window could be perfectly normal while not making sense in the context of a longer time horizon. For this reason, we believe that there are two types of semantic requirements of the representations and the latent views of LNT:

- *Contextualized semantics* (Addressed by $\mathcal{L}_{\text{DDCL}}$): views should reflect how the time window relates to the rest of the time series at different, longer time horizons, i.e. the similarity $h(z_t^{(l)}, W_k c_{t-k})$ is maximized for different $k$ and each $l$.

- *Local semantics* (Addressed by $\mathcal{L}_{\text{CPC}}$): views $z_t$ should share semantic information with the current time window $x_t$, which CPC achieves by maximizing its mutual information Oord et al. (2018).

Both loss contributions of LNT facilitate these requirements. CPC contributes local latent representations and context representations. The semantic content of the views is managed by the DDCL loss.

# 5 Experiments

For experimental evaluation of LNT in comparison to other methods, we study three challenging datasets. We first describe the datasets, baselines and implementation details. In Section 5.3, we present our findings: LNT outperforms many strong baselines in detecting anomalies in the operation of a water distribution and a water treatment system and accurately finds anomalies in speech. In Section 5.4, we provide visualizations of the local transformations that are learned by LNT. Finally, in Section 5.5 we analyze the performance of LNT in comparison to CPC based alternatives. Our findings that LNT is consistently superior, complements our theoretical analysis in Section 4 on why CPC and transformation learning should be combined.

## 5.1 Datasets

We evaluate LNT on three challenging real-world datasets, namely the Water Distribution Dataset (WaDi) Ahmed et al. (2017), the Secure Water Treatment Dataset (SWaT) (Goh et al., 2016) and the *Libri Speech Collection* (Panayotov et al., 2015). The first two datasets are provided with labeled anomalies in the test set. As recent observations in Wu & Keogh (2020) show, many popular datasets for time series AD seem to be mislabeled and flawed, which results in the revival of synthetic datasets Lai et al. (2021). The Libri Speech data is augmented with *realistic synthetic anomalies*.

**Water Distribution**    The dataset is acquired from a water distribution testbed and provides a model of a scaled-down version of a large water distribution network in a city (Ahmed et al., 2017). The time series data is 112-dimensional with readings from different sensors and actuators such as pumps and valves. The training data consists of 14 days of normal operation sampled with a frequency of 1 Hz, resulting in a series length of 1048571. The test set consists of 2 days of additional operation (172801 time steps), during which 15 attacks were staged with an average duration of $\approx 12$ minutes.

**Secure Water Treatment**    This dataset is from a testbed for water treatment (Mathur & Tippenhauer, 2016) that evaluates the Cyber Security of a fully functional plant with a six-stage process of filtration and chemical dosing. Goh et al. (2016) collected 11 days of operation data. Under normal operation 51 sensor channels are recorded for 7 days yielding a training time series of length 475200. For the test data of length 224960, 36 attacks were launched during the last 4 days of the collection process. As suggested in Goh et al. (2016); Li et al. (2019), the first 21600 samples from the training data are removed for training stability.

We follow the experimental setup of He & Zhao (2019) and take the first part of the collection under attack as the validation set and drop channels which are constant in both training and test set, yielding a time series of 45 dimension.

**Libri Speech**    The *LibriSpeech dataset* Panayotov et al. (2015) is an audio collection with spoken language recordings from 251 distinct speakers. We adopt the setup of Oord et al. (2018) with their train/test split and unsupervised training on the raw time signal without further pre-processing. For AD benchmarks, we randomly place additive pure sine tones of varying frequency (20 - 120 Hz) and length (512 - 4096 time steps) in the test data, yielding consecutive anomaly regions making up $\approx 10\%$ of the test data. Speech data offers a challenging benchmark for deep AD methods since speech typically exhibits complex temporal dynamics, due to high multi-modality introduced through different speakers and word sequences (Oord et al., 2018).

## 5.2 Baselines and Implementation Details

| Types | SWaT | WaDi | Libri |
|---|---|---|---|
| # neurons | 24 | 32 | 64 |
| # layers | 2 | 2 | 3 |
| activation | ReLU | ReLU | ReLU |
| bias | False | False | False |

Table 1: Neural Transformation Hyperparameters

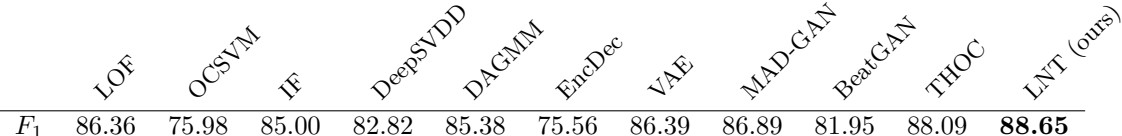

| | LOF | OCSVM | IF | DeepSVDD | DAGMM | EncDec | VAE | MAD-GAN | BeatGAN | THOC | LNT (ours) |
|---|---|---|---|---|---|---|---|---|---|---|---|
| $F_1$ | 86.36 | 75.98 | 85.00 | 82.82 | 85.38 | 75.56 | 86.39 | 86.89 | 81.95 | 88.09 | **88.65** |

Table 2: F1-scores (%) for the Secure Water Treatment Dataset (SWaT). Baseline results as reported in Shen et al. (2020).

**Baselines**  We study LNT in comparison to different classes of AD algorithms, ranging from classical methods to recent advances in deep AD. They include (i) classical methods, such as Isolation Forests (Liu et al., 2008), PCA reconstruction error (Shyu et al., 2003), and Feature Bagging (Lazarevic & Kumar, 2005), (ii) auto-regressive future predictions with LSTM (Hundman et al., 2018) and GDN (Deng & Hooi, 2021), which uses a graph to model the relations among variables as attention for the prediction, (iii) methods that estimate the density of the data, such as KNN (Angiulli & Pizzuti, 2002), LOF (Breunig et al., 2000), combinations with deep auto-encoders DAGMM (Zong et al., 2018), (iv) methods that employ a one-class objective, including OC-SVM (Schölkopf et al., 1999), DeepSVDD (Ruff et al., 2018) and THOC (Shen et al., 2020) for time-series, (v) methods that leverage the reconstruction of an auto-encoder with EncDec-AD (Malhotra et al., 2016) and LSTM-VAE (Park et al., 2018) (vi) and finally methods that use the ability of GANs to discriminate fake examples, like BeatGAN (Zhou et al., 2019) and MAD-GAN (Li et al., 2019).

**Implementation Details**  For LNT, the hyperparamaters are adopted from those reported by Oord et al. (2018) for CPC: especially $c_t \in \mathbb{R}^{256}$, $z_t \in \mathbb{R}^{512}$ and $K = 12$ for experiments with LibriSpeech data. The data is processed in sub-sequences of length 20480 for both training and testing. Since the other datasets contain way less diverse data points and show simpler temporal dynamics, the embeddings size, and thus the capacity of the model, is reduced to $c_t \in \mathbb{R}^{32}$, $z_t \in \mathbb{R}^{128}$. Also, the time-convolutional encoder network is down-sized to filters $(3, 3, 4, 2)$ and strides $(3, 3, 4, 2)$ resulting in the convolution of 72 time steps.

We consistently choose $L = 12$ distinct learned transformations $T_l(z_t)$ for all datasets. Each is represented by an *MLP* with properties summarized in table 1. The final layer always shares the dimensionality of $z_t$ and is applied as a *multiplicative mask* with *sigmoid* activation to it. Additional implementation details are in the appendix.

The crucial part of LNT in terms of hyperparameters is the representation learning with CPC. Its parameters depend on the frequency of observations and sequence lengths in the time series data at hand and can be determined as for any other representation learning. Here, the validation data does not need any anomalies in order to find good hyper-parameters. These preceding optimizations imply different sizes for the embedding vectors $z_t, c_t$ that depend on the size of and inherent variations contained in a dataset. Afterward, as a rule of thumb, the size of the neural transformations are just scaled proportional to these embedding sizes and validated with the (smaller) validation sets containing anomalies.

### 5.3  Results

We judge the anomaly scores predicted by the algorithms for each time step individually. Since the ratio of anomalies is imbalanced in the data, we evaluated the prediction performance with the $F_1$ score, consistent with previous work. Additionally, we also report results using the ROC curve. The area under the curve (ROC-AUC) is a metric to judge the quality of the anomaly score independent of the choice of threshold, which is specifically chosen for its additional insights beyond the evaluation of a single threshold.

The results on the SWaT and WaDi datasets can be seen in Tables 2 and 3a, respectively. The ROC curves of our method on the SWaT and WaDi datasets are provided in Figures 4a and 4b. For SWaT, our approach (LNT) outperformed a set of challenging baselines as reported by Shen et al. (2020) with the highest $F_1$ score (88.65%). Meanwhile for WaDi, our model produces comparable results both in terms of $F_1$ and precision, with the highest recall value[2]. Notably, GDN achieves the highest precision on WaDi even though our own run, *GDN (rerun)*, performed slightly worse than the reported results in Deng & Hooi (2021). When we

---

[2]In all experiments (and methods) the thresholds on the continuous anomaly score are optimized for the best $F1$.

| Method | $F_1$ | Prec | Rec |
|---|---|---|---|
| PCA | 0.10 | 39.53 | 5.63 |
| KNN | 0.08 | 7.76 | 7.75 |
| FB | 0.09 | 8.60 | 8.60 |
| EncDec-AD | 0.34 | 34.35 | 34.35 |
| DAGMM | 0.36 | 54.44 | 26.99 |
| LSMT-VAE | 0.25 | 87.79 | 14.45 |
| MAD-GAN | 0.37 | 41.44 | 33.92 |
| GDN | **0.57** | **97.50** | 40.19 |
| GDN (rerun)† | 0.47 | 83.76 | 33.06 |
| GDN (adj.) † | 0.38 | 29.38 | 54.22 |
| LNT (ours) † | 0.39 | 29.34 | **60.92** |

(a) Water Distribution Data (WaDi).

| Method | AUC | Prec | Rec | $F_1$ |
|---|---|---|---|---|
| LSTM † | 0.58 | 15.0 | 15.0 | 0.15 |
| THOC † | 0.82 | 30.2 | 30.0 | 0.30 |
| LNT (ours) † | **0.93** | **65.0** | **65.0** | **0.65** |

(b) Synthetic anomalies randomly placed in the LibriSpeech dataset.

Table 3: Experimental results for additional datasets. Baseline results are taken from Deng & Hooi (2021), except for the methods marked with † which are derived from our own experiments.

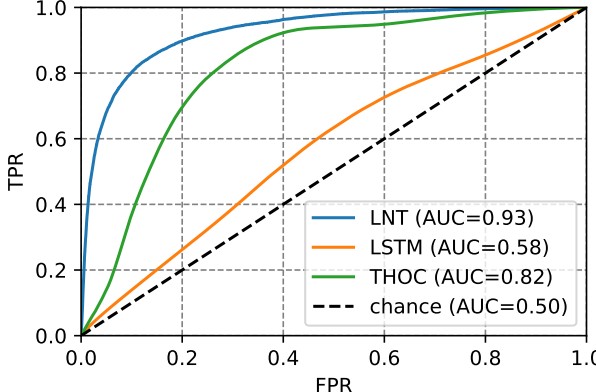

Figure 2: Our approach LNT outperforms deep baselines in AD on speech data in terms of ROC-AUC curves.

adjust the thresholds in *GDN (adj)* to have a comparable precision as LNT it has a lower recall (54.22%) than our method (60.92%). In many mission-critical applications, detecting as many anomalies as possible is often much more important, as a false negative can do more harm than a false positive. This makes the high recall of LNT (60.92%) preferable while retaining an acceptably high $F1$ score.

We argue that the novel criterion for AD based on contrasting learned latent data transformations allows LNT to also uncover some of the harder detectable anomalies in the dataset. A similar behaviour can also be observed for the LibriSpeech data with results in terms of ROC curves shown in Figure 2. Here, LNT clearly outperforms both deep learning methods. This shows that detecting anomalies within speech data with its complex temporal dynamics is indeed a challenging task for many deep AD algorithms. Especially the future predictions of LSTM perform only slightly better than random chance in this experiment for all possible thresholds. This emphasizes the benefit of contrasting of neural transformations to uncover such hard anomalies. Additional metrics for this experiment are reported in Table 3b.

## 5.4 Visualization of Transformations

In general, it is considered hard to get insights from embedding visualizations for $z_t$ in the latent space. Hence, to make the transformations interpretable in terms of semantics, we propose to visualize them in data space. We reuse the encoder as described in Section 3.1 and enrich it with a separate decoder. We train the decoder to reconstruct the (non-transformed) input data while freezing the encoder weights. The trained decoder is then applied to transformed embeddings to visualize them in data space.

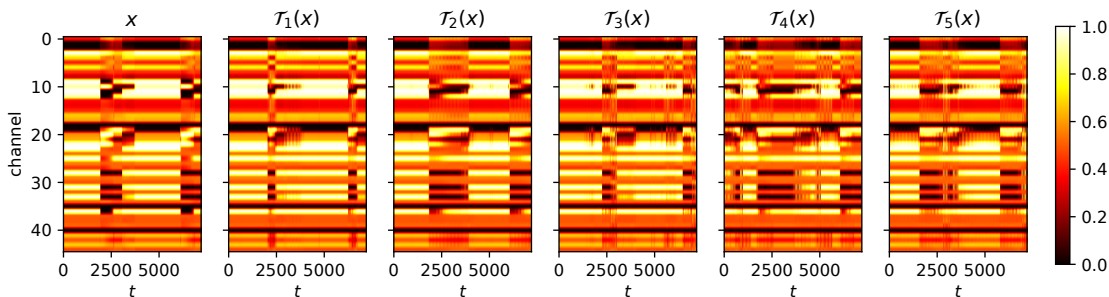

Figure 3: Visualizations of selected transformations in data-space that show semantically interpretable behaviour, such as altered delays in specific channels. Representations from SWaT dataset are decoded with a seperatly trained auto-encoder.

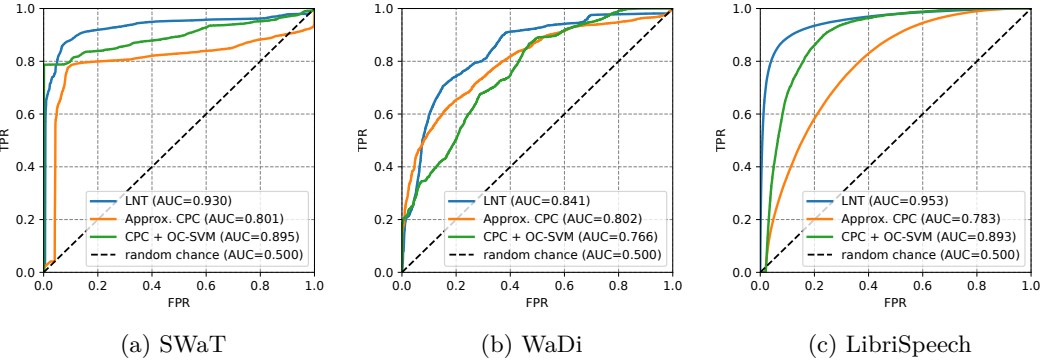

(a) SWaT          (b) WaDi          (c) LibriSpeech

Figure 4: Improvement of LNT over CPC scoring evaluated for different datasets. The combination of transformation learning with local representation learning of CPC consitently outperforms the other variants of CPC for anomaly scoring.

We chose a subset $\{\mathcal{T}_i\}_{i=1}^5$ of five transformations which showed interpretable behavior in experiments with *SWaT* as shown in Figure 3: For the non-transformed series $x$ the signal jumps in channels 25 and 36 at $t \approx 2500$. This jump is delayed for channels $26 - 35$. Interestingly, we found that this delay is altered by the learned transformations. For example, $\mathcal{T}_1$ removes this delay causing the signal jump for all aforementioned channels at $t \approx 2500$. In contrast, $\mathcal{T}_2$ affects the series oppositely by enlarging this delay.

In summary, these transformations produce *semantically* meaningful and *diverse* views of the time series. Admittedly, current interpretations are still rather high-level and fairly limited from application standpoints. However, without domain knowledge, there exists no *gold standard* for a good transformation on the data to compare against. This was the original motivation for the usage of *learnable transformations*, as effective data augmentation for AD.

### 5.5 Emperical Ablation Study

Recall, that we defined LNT to be a composition of CPC and *neural transformations* trained from a joint loss $\mathcal{L} = \mathcal{L}_{\mathrm{CPC}} + \lambda \cdot \mathcal{L}_{\mathrm{DDCL}}$. Theorem 1 provided a theoretical argument for the advantage of LNT over an approach purely based on DDCL. Also in practice this leeds to a solution with close to random performance in detecting anomalies and is thus not further considered in the following.

Instead, we study the reverse ablation: the advantage of LNT over pure CPC. There are several ways to use CPC to detect anomalies: (i) directly use the CPC-loss to score anomalies (de Haan & Löwe, 2021) or (ii) use CPC as a feature extractor and then run another AD method such as OC-SVM on the extracted features. One disadvantage of (i) are the negative samples. They make it nontrivial to evaluate the CPC-loss on

test data. We employ a practical implementation (Approx. CPC) without negative samples at test time. de Haan & Löwe (2021) argue that taking samples from the test data is biased and using the training data is infeasible in practice. In contrast, DDCL is deterministic and the alternative views are all constructed from a single sample. It is hence straightforward to use it to score anomalies at test time. From the results in Figure 4, we found that the combination of transformation learning with local representation learning of CPC consistently outperforms the considered variants of CPC for AD in all three datasets. This connects to the discussion about *contextualized semantics* in Section 4. Comparing LNT with CPC + OC-SVM supports our claim: While the OC-SVM with CPC input features has access only to the *local semantics* in the CPC representations, the performance of LNT in Figure 4 is consistently superior and can be explained by its transformations exhibiting both *contextualized semantics* and *diversity*.

## 6 Conclusion

We propose a novel self-supervised method, LNT, to detect anomalies within time series. The key ingredient is a novel training objective combining representation and transformation learning. We prove that both learning paradigms complement each other to avoid trivial solutions not appropriate for AD. We find in an empirical study that LNT learns to insert delays, which allows it to outperform many strong baselines on challenging detection tasks.

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

## Appendix

## A    Further Implementation Details

In this section the implementation details for the experiments conducted in the main paper are further elaborated. These include our method (LNT) as well as all baselines that we implemented for comparision.

### A.1    Hardware

All experiments were run on virtualized hardware with 8 CPU cores of type *Intel(R) Xeon(R) Gold 6150* running at 2.70 GHz, 32 GB RAM, and a single *TeslaV100-SXM2* with 32 GB of gpu memory. Consistently we use *Python 3.9*, *PyTorch* in version 1.8.1 with *CUDA* in version 11.1 and *cuDNN* in version 8.0.5.

### A.2    Hyperparameters

**LNT**    The hyper-parameters for our method were determined by the following procedure. Starting with the hyper-paramters as reported in Oord et al. (2018), the sizes of the embeddings $z_t$ and $c_t$, which also determines the number of memory units in the recurrent part $g_{\mathrm{ar}}$, and the number of parameters in the convolutional encoder $g_{\mathrm{enc}}$ are downsized to fit the complexity and amount of data in the other datasets. To find a well generalizing setup, a hold-out validation set (split from the training data) was used. For Libri-Speech we considered the hyper-parameters as optimal and didn't change them. As a rule of thump, the sequence length for training and the width of the strided temporal convolutions were always chosen in a way such that the number of recurrent steps $g_{\mathrm{ar}}$ takes matches with the setup $(= 128)$ in Oord et al. (2018).

LNT is trained for 100 epochs, respectively 500 epochs on SWaT and WaDi, with learning rate $2 \cdot 10^{-4}$, batch size 32 and $\lambda = 10^{-3}$.

### A.3    Baselines in LibriSpeech Experiments

The following hyperparameter setups are used for the experiments conducted with synthetic anomalies in LibriSpeech data.

**LSTM**    Here, a standard Long Short Term Memory (LSTM) network with 2 layers and 256 hidden units each was chosen. With this setup the number of hidden units aligns with the LNT setup and the multiple layers should account for the missing encoder structure in LSTM. It is trained until convergence, which took approximately 100 epochs, with batch size 32, learning rate $2 \cdot 10^{-4}$ and a dropout of 0.3.

**THOC**    Here, the Implementation was kindly provided by the authors. We used a smaller sub-sequence length of 1024 for training due to the high memory load of the model. Predictions at test time are stitched together to align with the longer sequence length. The method is trained to fit 3 layers hierarchical with dilations $(1, 2, 4)$, 128 hidden units and 6 clusters in each layer. The method is trained with learning rate $10^{-3}$ and batch size 32 and converged after 50 epochs.

# B  Notation Details

The following table summarizes the notations used in the main paper.

| Notation | Description |
|---|---|
| $x_t$ | patch of (multivariate) measurements of a time series $\mathbf{x}$ in the time interval $[t - \tau, t + \tau]$ for a fixed window size $\tau$ |
| $z_t$ | *local representation* $z_t = g_{\mathrm{enc}}(x_t)$ of a time series patch $x_t$ produced by the encoder $g_{\mathrm{enc}}$ |
| $c_t$ | *context representation* $c_t = g_{\mathrm{ar}}(z_{\leq t})$ that summarize the history of local representations $z_{\leq t} := z_{0:t}$ with an autoregressive network $g_{\mathrm{ar}}$ |
| $W_k$ | matrix to linearly predict embeddings $k$ steps into the future |
| $W_k c_t$ | linear (future) prediction of the ground truth embedding $z_{t+k}$ |
| $\mathcal{T}_l(\cdot)$ | a neural transformation (i.e. a neural network) with parameters $\theta_l$ |
| $z_t^{(l)}$ | a *latent view* $z_t^{(l)} = \mathcal{T}_l(z_t)$ of a local representation $z_t$ at time $t$ acquired by applying transformation $\mathcal{T}_l$ |
| $\ell_t^{(k,l)}$ | the contribution to the DDCL loss for a specific transformation $l$ and $k$-step future predictions with $W_k$ |
| $\ell_t(x_{\leq t})$ | all DDCL loss contributions up to a time step $t$; used as the *anomaly score* for that time step |
| $h(\cdot, \cdot)$ | exponated cosine similarity between embeddings $h(z_i, z_j) := \exp \frac{z_i^T z_j}{\|z_i\|\|z_j\|}$ |

Table 4: Overview of the notation used in the paper

