# OpenReview forum: "Detecting Anomalies within Time Series using Local Neural Transformations"
_TMLR — Rejected by TMLR_

### Review · Reviewer_eT2k · 2023-01-04

**Summary Of Contributions:**

This paper develops a new method for detecting anomalies in time series data. The proposed method borrows ideas from self-supervised (contrastive) learning and self-supervised anomaly detection and applies them in the context of time series analysis. The proposed method (Local Neural Transformations or LNT) is an encoder-decoder architecture trained using a novel objective function with two main components (both based on contrastive losses):

* representation learning: a standard contrastive predictive coding loss that is used to learn the time-series dynamics.

* transformation learning: a novel dynamic deterministic contrastive loss (DDCL) that contrasts different latent views of each time step while covering different sequence lengths and time separations between views.

The authors show that the objective function is theoretically sensible as the model will collapse to trivial solutions if the representation loss is not included. They show empirically that the proposed method outperforms previous methods in various real-world datasets.

**Audience:**

Yes

**Broader Impact Concerns:**

I do not believe that this paper requires a broader impact statement.

**Claims And Evidence:**

Yes

**Requested Changes:**

1- Formally defining temporal anomalies.

2- Adding ablated versions of the method with CPC and DDCL removed or replaced with other variants (Tables 2 and 3).

3- Extending the analysis on learned temporal transformations in Sec 5.4 to multiple data sets.

4- Adding confidence intervals to all reported metrics.

**Strengths And Weaknesses:**

Strengths

(1) Problems related to time-series data are not as extensively researched as in imaging and language modalities. This paper addresses anomaly detection in time-series data and develops methods that are tailored to the peculiar and interesting aspects of time-series (temporal correlations, window-length of series history, etc), as well as discovering interesting facts about transformations/data augmentations that are useful in the context of time series (e.g., insertion of delays).

(2) The proposed method relies on learned transformations rather than hand-crafted ones, which is a much more sensible choice particularly in non-imaging setups where handcrafted augmentations may not be straightforward to design.

(3) The paper has a nice balance between empirical evidence and theoretical backing.

Weakness

(1) The paper never defines what an anomaly is within the context of time-series data. Unlike in tabular or imaging data, a time-series entails repeated observations and predictions, and one can define anomalous observations as entire sequences or individual observations. It seems from the proposed method that the authors define an anomaly as an "out of distribution" observation conditioned on the observed history at all time steps. But it is not clear if this is a good definition of anomalies as in this case, change points or non-stationary dynamics may wrongly be perceived as anomalies.

(2) The theory focuses on the triviality of models learned via the DDCL loss only (with no CPC loss). I think that this result is secondary in the context of the original focus of the problem. I expected the theoretical analysis to prove that the proposed loss is strictly larger for anomalous observations (given a formal definition of anomalies) similar to the results in the one-class classification literature.

(3) The experiments did not compare ablated versions of the proposed method, which I think is important given that the method has multiple loss components and architectural choices.

(4) The analysis on interpretability of learned transformations (Sec 5.4) seems a bit ad-hoc and immature. I think the discovered transformation will be more interesting if the authors can reproduce them across multiple data sets instead of showing it for SWaT only.

(5) All reported metrics had no standard errors/CIs. I think this is important given that F-1, precision and recall can have very high variances especially that anomalies are rare events by definition.

---

> ### Author Response · Authors · 2023-02-15
> **Response**
>
> Dear reviewer,
>
> thank you for your insightful comments.
>
> You asked for:
> - Formal Definition of Temporal Anomalies along with proof that the loss is strictly larger for anomalous observations
> 	- We added a  definition of a temporal anomaly to the paper. It is close to what you already inferred from the context. Even though we respectfully disagree with your comment on the false detection of all change points: As long as the change points that are considered normal are sufficiently covered within the training data the model should be able to incorporate this into its learned representations and thus not alert an anomaly. Please consider the LibriSpeech data: it contains a variety of speakers and different spoken sentences (i.e. change points in the patterns of oral sounds). These are highly unsteady time series. Nevertheless, almost only our synthetic anomalies are alerted.
> 	- Regarding the proof of correctness: Such proof is hard to provide since the method involves deep representation learning and performs operations in a latent space. Related work with a similar approach (eg. THOC, DeepSVDD) can not provide such proof either for the very same reason. Future work may focus on that open research question but it is out of scope for this work.
> - An ablation study
> 	- Please note, that the paper already contained (sec. 5.5) the ablation study that you requested. We reworked the introductions to the corresponding sections to better emphasize the purpose of our experiments. In our ablation study, we considered different approaches to use CPC (i.e. DDCL was ablated) for anomaly detection directly. The reverse ablation (DDCL without CPC) is discussed in our theoretical analysis (sec. 4.1) - with the result that the solution collapses to something completely useless for anomaly detection (which also holds practically true).
> - Better Visualization of the learned transformations
> 	- We are no experts in any of the domains making more insightful interpretations of the learned transformations challenging.
> - standard errors and confidence intervals
> 	- The most related work and source of the baselines ([Shen et. al 2020](https://proceedings.neurips.cc/paper/2020/file/97e401a02082021fd24957f852e0e475-Paper.pdf)) does not provide standard errors and we adopted their experimental setup.
>
> We uploaded a **revised version** of the paper with all changes marked in blue.
>
> Kind regards,
>
> the anonymous authors

---

### Review · Reviewer_4pVV · 2023-01-05

**Summary Of Contributions:**

This paper proposes a method for time series anomaly detection, which unifies time series representations with a novel approach for learning local transformations. Theoretical analysis proves that both learning paradigms complement each other to avoid trivial solutions which are not appropriate for detecting anomalies.


**Audience:**

Yes

**Broader Impact Concerns:**

No.

**Claims And Evidence:**

No

**Requested Changes:**

1. The authors could provide more comparisons with the existing self-supervised anomaly detection works for time series [2][4].

2. The authors could provide more theoretical analysis of the proposed method, i.e., the complexity of the proposed work.

3. Please compare the proposed method with some recent works, e.g. [5], in the experiment.

4. Clarify the contributions.


Minor:
1.	In the introduction, “there is much less work on other domains such as time series” this is inaccurate.
2.	I suggest the authors provide a table to describe all the notations used in this paper.

I would appreciate it if the authors can address these concerns.



**Strengths And Weaknesses:**

This paper proposes a method for time series anomaly detection with some theoretical guarantee. The proposed method is interesting. Although the provided theoretical analysis is limited but practical. And the experiments are carried out on three challenging real-world datasets to evaluate the proposed method, which is limited but promising. However, I have some concerns as follows.

1. In the Introduction, the authors mentioned ``While AD has been an important field in machine learning for several decades, promising performance gains have been primarily reported in applying deep learning methods to high-dimensional data such as images. With few exceptions, there is much less work on other domains such as time series.’’ This is inaccurate. Time series anomaly detection has been widely studied in recent years, please refer to [1][2][3][4][5]. Related works in this direction should be discussed.

2. Can the authors compare the proposed method with some self-supervised anomaly detection works for time series, e.g, [2][4]. I would appreciate it if the author can make some comparisons.

3. I suggest the authors provide a table that describes all the notations used in this paper. I would appreciate it if the authors can provide more theoretical analysis of this work, e.g, the complexity of the proposed method since the existing theoretical analysis is somewhat weak.

4. This work seems to simply combine some self-supervised methods with some deep learning architectures for time series anomaly detection. I encourage the authors to clearly clarify their contributions.

5. In the experiments, I encourage the authors to compare the proposed method with some recent works, e.g., [5].


Reference

[1] BeatGAN: Anomalous Rhythm Detection using Adversarially Generated Time Series. IJCAI. 2019.

[2] TimeAutoAD: Autonomous Anomaly Detection with Self-Supervised Contrastive Loss for Multivariate Time Series. IEEE Transactions on Network Science and Engineering.

[3] TAnoGAN: Time series anomaly detection with generative adversarial networks.  IEEE Symposium Series on Computational Intelligence (SSCI).

[4] Self-Supervised Learning for Time-Series Anomaly Detection in Industrial Internet of Things.

[5] Graph neural network-based anomaly detection in multivariate time series. AAAI, 2021.

[6] A Deep Neural Network for Unsupervised Anomaly Detection and Diagnosis in Multivariate Time Series Data. AAAI 2019

---

> ### Author Response · Authors · 2023-02-15
> **Response**
>
> Dear reviewer,
>
> thank you for your insightful comments.
>
> You asked for:
> - Add missing related work in the introduction
> 	- We changed the corresponding paragraph in the introduction according to your suggestion. Thanks for pointing that out.
> - Analysis of the Complexity
> 	- We added an analysis of runtime complexity for both the training and prediction phases. Please see the new section 4.2.
> - Clarification of the contribution: a simple combination of deep learning architectures
> 	- We respectfully disagree. Anomaly detection with data transformation shows superior performance for images and also tabular data (compare Qiu et. al. 2021). With this work we try to address the open research question on how to effectively extend these ideas to anomaly detection within time series? Our theoretical analysis shows that it needs to be combined with a temporal representation learning (i.e. CPC) to avoid a trivial collapse. Our empirical ablation (sec. 5.5) on the other hand shows that DDCL adds additional value in terms of anomaly detection performance to pure CPC.
> - table with all notations
> 	- We added such a table to the appendix. Thanks for the suggestions.
>
> We uploaded a **revised version** of the paper with all changes marked in blue.
>
> Kind regards,
>
> the anonymous authors

---

### Review · Reviewer_8JWg · 2023-02-01

**Summary Of Contributions:**

The paper proposes an unsupervised local anomaly detection method for time series data.
The proposed method defines an anomaly score for time series by combining the existing Contrastive Predictive Coding (CPC) loss with a novel Dynamic Deterministic Contrastive Loss (DDCL). The main idea of the proposed DDCL loss is to learn local transformations of time series representations. Different transformed representations should be diverse but at the same time predictive of the surrounding context.

The proposed approach has theoretical advantages over existing methods (requires no labeled anomalies, is deterministic). Empirically, the proposed method shows more accurate anomaly compared to existing works, as demonstrated by experiments on 3 real-world datasets with synthetic anomalies.

**Audience:**

Yes

**Claims And Evidence:**

Yes

**Requested Changes:**

There is one question that I would like to see answered during the discussion period and clarified in the paper.

How important is the choice of the model hyperparameters for the proposed approach? As shown in Table 1, a different hyperparameter setting is used for every dataset considered in the experiments.

- Do the qualitative conclusions of the paper still stand if we instead trained the model with different hyperparameters (e.g., same setting for all datasets)?
- How much variance does the proposed approach exhibit given different model configurations/random initializations?

The specific choice in the paper is justified by the "complex temporal dynamics" of the Libri dataset. However, it's unclear, how these parameters should be determined in practice given the absence of labeled anomalous data that could be used to tune the models.

**Strengths And Weaknesses:**

Strengths:
- The literature review is thorough, and the contribution of this paper is clearly positioned with respect to related works.
- The proposed approach is conceptually simple and demonstrates a consistent improvement over existing anomaly detection methods for time series.
- Some aspects of the proposed method are motivated  by a theoretical analysis
- The paper is well-written and is easy to follow.

Weaknesses:
- Some aspects of the experimental setup:
	- The empirical evaluation is limited to 3 datasets, and not all baseline methods are tested on all datasets.
	- It would be beneficial to include simple baseline methods, such as those mentioned in [(Wu & Keogh)](https://arxiv.org/abs/2009.13807)
	- Hyperparameter choices and their effect on model's accuracy could be explained more clearly (see Requested Changes below).

---

> ### Author Response · Authors · 2023-02-15
> **Response**
>
> Dear reviewer,
>
> thank you for your insightful comments.
>
> You asked for:
> - Do the qualitative conclusions of the paper still stand if we instead trained the model with different hyperparameters (e.g., same setting for all datasets)? However, it's unclear, how these parameters should be determined in practice given the absence of labeled anomalous data that could be used to tune the models.
> 	- The crucial part of LNT in terms of hyperparameters is the representation learning with CPC. Its parameters depend on the frequency of observations and sequence lengths in the time series data at hand and can be determined as for any other representation learning. Here, selecting the wrong set of hyper-parameters can cause the performance to completely collapse. But, the validation data does not need any anomalies in order to find good hyper-parameters to that end. These preceding optimizations imply different sizes for the embedding vectors z and c that depend on the size of and inherent variations contained in a dataset. Afterward, as a rule of thumb, the size of the neural transformations are just scaled proportional to these embedding sizes and validated with the (smaller) validation sets containing anomalies. Again, the transformation sizes need to be optimized but it does not completely collapse with suboptimal parameters if the underlying representation learning is converging. We extended the corresponding hyper-parameter section in the paper accordingly.
> - It would be beneficial to include simple baseline methods, such as those mentioned in [(Wu & Keogh)](https://arxiv.org/abs/2009.13807)
> 	- Work by Wu/Keogh addresses the issues of (i) triviality, (ii) unrealistic anomaly density, (iii) mislabeled ground truth, and (iv) run-to-failure bias in existing anomaly detection benchmarks. Our synthetic experiments on LibriSpeech explicitly address the concerns (ii)-(iv) by having control over the process of generating artificial anomalies. Specifically, we set only 10% of a time series to be an anomaly to have a realistic density. Mislabeled ground truth can be excluded for synthetic anomalies as well and since we equally and randomly place anomalies within a time series in order to not have any bias in the position. Additionally, we chose the other datasets for the reason of not being on the list of Wu/Keogh. The only concern left is (i) triviality. Here, we would like to emphasize that we are pursuing a self-supervised anomaly detection approach. This means that an algorithm needs to learn a robust profile of normality without having a specific anomaly in mind. The simple rules derived by Wu/Keogh always contain magic numbers that need to be tuned for a specific anomaly at hand. This is a supervised approach in our perspective. (With the human intuition for good rules replacing a representation learning approach) Detecting anomalies might be always easy - in the hindsight, once you know for the kind of anomalies you are looking for. Learning a profile of normality on the other hand is a challenging representation learning task. Nevertheless, we could add such a baseline: for the LibriSpeech signal the moving mean should be zero anyway, no matter whether some additional sine tone is added or not.
>
> We uploaded a **revised version** of the paper with all changes marked in blue.
>
> Kind regards,
>
> the anonymous authors

---

### Review · Reviewer_obvq · 2023-02-06

**Summary Of Contributions:**

The paper proposes a self-supervised time series anomaly detection technique, labeling individual time points as anomalous or not without access to labeled training data. The proposed self-supervised loss combines contrastive predictive coding (CPC) and (a modified version of) the deterministic contrastive loss (DCL), by applying learnable transformation to representations of time series windows and encouraging the transformed representations to be both predictive of future representations as well as dissimilar to each other. The paper shows that the proposed DDCL loss by itself can lead to collapse to a trivial constant solution, which is prevented by combining it with CPC. The method is evaluated on two datasets with real anomalies (WaDi and SWaT), as well as one dataset (LibriSpeech) augmented with synthetic anomalies (added pure sine waves).

**Audience:**

Yes

**Broader Impact Concerns:**

No concerns around the ethical implications (other than those which would apply to any anomaly detection technique).

**Claims And Evidence:**

No

**Requested Changes:**

* [critical] Expand the quantitative empirical evaluation. For SWaT, include stronger competitors. For WaDi, the question around the performance of GDN needs to be resolved. If an argument around the tradeoff between precision and recall is to be made, the threshold needs to be chosen such that either precision or recall are fixed, in order to make the results comparable. Additional datasets and/or experiments with other ways of injecting synthetic anomalies into LibreSpeech would further strengthed the paper.
* It is not clear how exactly training examples are sampled and how mini-batches are constructed. Is there a difference in how this is done for the different loss components (CTC and DDCL)? This should be clarified.
* The paragraph after the proof in section 4, providing intuition about how the different loss components contribute, could be clarified. Additionally, it would be good to have experimental support for the claims made here.
* The visualization of the transformations is nice, but could be illustrated and described more clearly.

**Strengths And Weaknesses:**

Strengths:
* The paper tackles a relevant problem with an interesting and novel approach.
* The writing is mostly clear and easy to follow.
* The paper provides support for its claims using theory (showing that DDCL by itself is not sufficient) as well as quantitative and qualitative empirical studies.

Weaknesses:
* The theoretical contribution is somewhat limited: While the theorem shows that the proposed DDCL loss has a trivial constant solution, which can be mitigated by adding the CPC loss as a regularizer, no other properties of this combined loss are elucidated.
* The empirical evaluation is not convincing: On SWaT, the method is shown to (slightly) outperform the chosen baseline. However, prior work has reported significantly higher performance on this dataset (e.g. F1 of 95.28 reported by Carmona et al., which is cited as related work). On WaDi, the baseline results are taken from (Deng & Hooi, 2021). However, in footnote 3 the authors call the validity of these results (which significantly outperform the proposed approach) into question, which makes this entire comparison somewhat dubious. On LibreSpeech, the authors add synthetic anomalies in the form of pure sine waves (questionable whether these can be called "realistic") and compare only to a weak LSTM baseline and THOC.

---

> ### Author Response · Authors · 2023-02-08
> **Question: Additional theoretical properties of the combined loss**
>
> Dear reviewer,
>
> thank you for your insightful comments. We are working on their incorporation.
> Can you please further elaborate on which additional theoretical properties of the combined loss you would like to see proven?
>
> The motivation behind the theorem is a two-fold ablation study for the combined loss (CPC + DDCL). Our theory investigates DDCL without CPC and provides proof of why this ablation renders the method useless. This finding (which has a direct practical impact, because methods trained on this loss are indeed useless) is accompanied by the reverse (empirical) ablation for CPC without DDCL which investigates different ideas on how to use CPC for anomaly detection directly (compare section 5.5). Again, the finding is that it is best to combine both losses.
>
> We will rename the section to "ablation study" and rework it to improve clarity regarding that aspect.
>
> Kind regards,
>
> the anonymous authors

---

> ### Author Response · Authors · 2023-02-15
> **Response**
>
> Dear reviewer,
>
> thank you for your insightful comments.
>
> You asked for:
> - The theoretical contribution is somewhat limited
> 	- We added an analysis of runtime complexity for both the training and prediction phases. Please see the new section 4.2. **This is also a property of the loss** since the number of representations z and inner products thereof determines the complexity of our method.
> - The question around the performance of GDN needs to be resolved
> 	- We addressed the problem by adding the performance of our own run of GDN to the table with the results for comparison. As you suggested, we also added an additional baseline with an adjusted threshold to match the precision level of LNT. We now compare the recall of LNT against this baseline. We agree with you that this was necessary to make the argument sound.
> - It is not clear how exactly training examples are sampled and how mini-batches are constructed
> 	- We added/changed the passages describing the mini-batch construction for each subsection in the method section.
> - Change the paragraph after the proof in section 4
> 	-  We reworked the corresponding section.
>
> We uploaded a **revised version** of the paper with all changes marked in blue.
>
> Kind regards,
>
> the anonymous authors

---

### Decision · Action_Editors · 2023-04-17

**Recommendation:** Reject

**Comment:**

The authors addressed critical pieces of feedback and beefed up the experimental section, but a number of questions raised by the reviewers remained unaddressed. For instance, the authors need to provide more compelling evidence about the validity of the results as a function of the hyperparameters and they need to better justify the reasons for not including baselines suggested by the reviewers.

Though the work is rejected at the moment, the authors are encouraged to submit a major revision of the submission that addresses these concerns.

**Audience:**

The work is of interest to a relatively narrow audience of the TMLR community. This being said time series is an important topic with a broad class of applications.

**Claims And Evidence:**

The claims made in the submission are reasonably supported and complemented by a limited theoretical analysis. One major concern, was the weak experimental section. The authors run additional baselines as requested, but did not include a number of baselines identified by the reviewers (such as TimeAutoAD), nor did they justify this. Hence, the reviewers remained unconvinced of the experimental validation post discussion. The reviewers were happy with the complexity analysis provided by the authors as well as the additional details to clarify areas of confusion they raised in the text.